# The Prognostic Value of Non-Invasive Ventilation in Patients with Acute Heart Failure

**DOI:** 10.3390/biomedicines13081844

**Published:** 2025-07-29

**Authors:** Pietro Scicchitano, Assunta Cinelli, Gaetano Citarelli, Anna Livrieri, Cosimo Campanella, Micaela De Palo, Pasquale Caldarola, Marco Matteo Ciccone, Francesco Massari

**Affiliations:** 1Cardiology Section, Hospital “F. Perinei”, 70022 Altamura, Italy; anna.livrieri693@gmail.com (A.L.); francesco.massari@asl.bari.it (F.M.); 2Emergency Unit, Hospital “F. Perinei”, 70022 Bari, Italy; assunta.cinelli87@gmail.com; 3Cardiology Section, Hospital “S. Paolo”, 70132 Bari, Italy; gaetanocitarelli8@gmail.com (G.C.); cosimocamp@gmail.com (C.C.); pascald1506@gmail.com (P.C.); 4Cardiac Surgery Unit, University of Bari, 70126 Bari, Italy; micaela.depalo85@gmail.com; 5Cardiology Section, University of Bari, 70126 Bari, Italy; marcomatteo.ciccone@uniba.it

**Keywords:** acute heart failure, prognosis, non-invasive ventilation, length of hospital stay

## Abstract

**Objectives:** Patients with acute heart failure (AHF) often receive initial non-invasive ventilation (NIV). This study aimed to evaluate the prognostic role of NIV in patients hospitalized for AHF. **Methods:** This was a retrospective cohort study. We enrolled patients admitted to our cardiac intensive care unit with a diagnosis of AHF. Anthropometric, clinical, pharmacological, and instrumental assessments were collected. Both in-hospital and 180-day post-discharge mortality were evaluated. **Results:** Among 200 patients (mean age 81 ± 9 years; 52% male), NIV was applied in 80 cases (40%). These patients had more severe NYHA functional class, a higher prevalence of de novo AHF, required higher diuretic doses, and had longer hospital stays. In multivariate analysis, NIV remained significantly associated with length of stay (LOS) (r = 0.26; *p* = 0.0004). In-hospital mortality was 5% overall and significantly higher in the NIV group compared to non-NIV patients (10% vs. 1.6%, *p* < 0.001). At 180 days, mortality was also significantly higher in the NIV group [hazard ratio (HR) 1.84; 95% confidence interval (CI): 1.18–2.85; *p* = 0.006]. After adjusting for age, BNP, CRP, arterial blood gas parameters, renal function, and LVEF, NIV remained an independent predictor of 180-day mortality (HR 1.61; 95% CI: 1.01–2.54; *p* = 0.04). **Conclusions:** Patients with AHF who required NIV exhibited more severe disease and longer hospital stays. NIV use was independently associated with both in-hospital and post-discharge mortality, suggesting its potential role as a prognostic marker in AHF.

## 1. Introduction

Acute heart failure (AHF) remains a leading cause of hospitalizations worldwide, imposing significant morbidity, mortality, and healthcare costs [1]. The clinical management of AHF is complex due to the heterogeneity of its presentation, ranging from mild congestion to life-threatening pulmonary edema [2]. Rapid stabilization of respiratory distress is critical for reducing complications and improving patient outcomes [3].

Non-invasive ventilation (NIV)—including continuous positive airway pressure (CPAP) or bilevel positive airway pressure (BiPAP)—may play a pivotal role in the general management of these patients [4]. NIV treatment decreases breathing effort, reduces ventricular preload and afterload, and improves dyspnea, blood oxygenation, and hypercapnia, as well as reducing the need for invasive mechanical ventilation [5].

These beneficial effects may explain the increasing use of NIV in AHF [6]. However, uncertainties still persist about the prognostic impact of NIV in the setting of AHF. The 3CPO trial—one of the largest randomized controlled trials on NIV in AHF—did not report any improvement in in-hospital mortality despite prompt amelioration of pulmonary discomfort [7]. A recent Cochrane analysis by Berbenetz et al. [8] found a reduction in in-hospital mortality in patients treated with NIV compared to standard oxygen therapy in cases of cardiogenic shock, although no impact was observed on hospital length of stay (LOS).

Conversely, a retrospective study of more than six million hospitalizations for AHF found a two-fold increase in in-hospital mortality risk in patients who underwent NIV treatment [6]. Older age, hypotension, and acute coronary syndrome may be considered possible determinants of in-hospital mortality in patients with AHF receiving NIV therapy [9].

Beyond the contrasting results on in-hospital mortality, the impact of NIV on post-discharge prognosis in patients hospitalized for AHF remains a matter of debate.

Nevertheless, the literature is scant regarding the prognostic value of NIV use in AHF patients after discharge [10].

The aim of this original article was to explore the possible association between the use of NIV in AHF and both LOS and post-discharge mortality, in order to clarify its role not only as an acute intervention but also as a predictor of disease trajectory.

## 2. Materials and Methods

### 2.1. Study Patients

This was a retrospective study. We consecutively enrolled patients who were hospitalized in the cardiac intensive care unit (ICU) due to AHF between January 2020 and December 2023.

The main inclusion criteria were hospitalization for AHF and age >18 years. The exclusion criteria were pneumonia/bronchitis, myocarditis, pulmonary embolism, acute coronary syndrome, recent or scheduled cardiac surgery, coronavirus disease (COVID-19), history of active neoplasms, previous heart transplantation, Use of invasive mechanical ventilation, and Haemodialysis. The flow chart of the study is included as Figure 1.

The primary variable of interest was the use of NIV (CPAP or BiPAP) within the first 24 h from admission.

Patients’ clinical and anthropometric baseline characteristics, underlying diseases and comorbidities, laboratory examinations, pharmacological treatments/background, and arterial blood gas analyses on room air at admission were collected.

AHF was defined as acute decompensation of chronic heart failure (ADHF) or “de novo” acute heart failure (de novo AHF). The left ventricular ejection fraction (LVEF) was calculated by echocardiography using Simpson’s method.

Brain natriuretic peptide (BNP) and high-sensitivity C-reactive protein (hsCRP) were assessed by a clinical chemistry analyser (Beckman Coulter AU680, Brea, CA, USA). Estimated glomerular filtration rate (eGFR) was calculated using the 4-variable Modified Diet in Renal Disease formula.

We assessed congestion status by considering its different components: BNP as a surrogate marker of “hemodynamic” congestion, estimated plasma volume status (ePVS, dL/g) as an expression of “intravascular” congestion, and hydration index (HI, %) and bioimpedance vector analysis (BIVA) as indicators of “peripheral” hydration status [11].

More specifically, ePVS was calculated using Duarte’s formula: [(1 − hematocrit)/(hemoglobin)]. HI was assessed at admission using BIVA (Bodygram 1.4, Akern RJL Systems, Florence, Italy), as previously reported [11].

The endpoint was in-hospital mortality and 180-day mortality, the latter assessed via medical records or national death records.

The study complied with the Declaration of Helsinki and was approved by the local Institutional Review Board. Written informed consent was obtained from each patient at inclusion (protocol no. 0081801/CE—29 October 2015, study number: 4816).

### 2.2. Statistical Analysis

Normally distributed variables were expressed as mean ± standard deviation, while non-normally distributed continuous variables were expressed as median and 95% confidence interval (CI). Differences between the two groups were compared using Student’s *t*-test or Mann–Whitney U test, as appropriate. Categorical variables were presented as percentages (%) and compared using the χ^2^ test.

A multivariate linear regression model was performed to assess the determinants of LOS. Multivariate logistic regression analysis was used to identify the predictors of NIV treatment. The trend of outcomes was expressed as Kaplan–Meier cumulative survival plots with Log-rank significance. The impact of NIV on the odds ratio of in-hospital and 180-day follow-up mortality was assessed by univariate logistic regression. Univariate and multivariate Cox proportional hazards regression models, with estimations of hazard ratios (HR) and 95% CI, were used to evaluate the association of variables with mortality.

*p*-values < 0.05 were considered statistically significant. Analyses were performed using STATA software, version 12 (StataCorp, College Station, TX, USA).

## 3. Results

Two-hundred patients were finally enrolled. Eighty (40%) patients received NIV within the first 24 h from the admission. Baseline characteristics of the study population and comparison between NIV and non-NIV groups are reported in Table 1.

The prevalence of pulmonary crackles was higher in patients on NIV, while other signs of clinical congestion (i.e., jugular venous distention and peripheral edema) as well as hemodynamic, intravascular, and peripheral biomarkers of congestion (i.e., BNP, ePVS, and HI) did not differ between the two groups. NIV treatment was mostly adopted in “de novo” AHF patients (Table 1) and in those with advanced New York Heart Association (NYHA) class (Figure 2).

Higher plasma levels in hsCRP and blood urea nitrogen (BUN) were also observed in NIV patients, as well as higher doses in intravenous furosemide. According to blood gas analysis, the NIV group showed higher partial pressure of arterial carbon dioxide (PaCO_2_, mmHg) and lower partial pressure of arterial oxygen (PaO_2_, mmHg) and oxygen blood saturation (Table 1).

Therefore, we tried to evaluate the predictors for NIV use in our population. The multivariate logistic regression analysis identified “de novo” AHF, NYHA functional class, furosemide doses, oxygen saturation, and PaCO_2_ as predictors, the C-Index set at 0.86 (Table 2).


The LOS was significantly higher in the NIV group (Figure 3) even after adjusting for confounding factors (Table 3).

Specifically, NIV treatment predicted prolonged hospitalization periods higher than 7 days with an odds ratio (OR) equal to 4.9 (95% CI 2.2–10.1).

The cumulative in-hospital mortality was 5%. It occurred after 9.0 days (95% CI: 2.4–17.1, days) in 10% of patients with NIV and in 1.6% of patients who did not undergo NIV-treatment (Figure 4A).

During the follow-up, the cumulative mortality was 11% at 30 days, 29% at 90 days, and 41% at 180 days. The prognostic value of NIV still persisted during the 180-day follow-up, but the performance and risk ratio progressively decreased (Figure 4B).

In our univariate Cox regression analysis, BUN, eGFR, BNP, hsCPR, and NIV treatment were predictors of mortality at 180-day follow-up (Table 4).

The Cox proportional hazard regression analysis rather demonstrated that the use of NIV remained significantly associated with mortality, along with BNP and BUN (Table 4).

## 4. Discussion

The role of NIV in the setting of AHF has been long debated, although definite data are lacking. Our study tried to evaluate the prognostic impact of NIV use in patients with AHF as compared to those who did not require it. The main results were the following: (1) NIV treatment was more frequently performed in patients with respiratory distress, more impaired functional NYHA class, “de novo” AHF, and those on higher dose of intravenous furosemide; (2) longer LOS was necessary in patients on NIV, independently from any other clinical and laboratory parameters; (3) the use of NIV might be considered as an independent predictor for in-hospital and 180-day follow-up all-cause mortality.

The efficacy of NIV in counteracting pulmonary edema was known since the early 1930s [12], as it represented a useful technique for reducing the congestion in purely cardiogenic AHF.

International guidelines [4,13] clearly recommend the use of oxygen only in those patients with SpO_2_ < 90% or PaO_2_ < 60 mmHg to correct hypoxaemia (class of recommendation I, level of evidence C), while suggesting non-invasive positive pressure ventilation in those with respiratory distress (respiratory rate >25 breaths/min, SpO_2_ < 90%) [class of recommendation IIa, level of evidence B] in order to ameliorate respiratory distress and reduce the need for invasive ventilation.

A recent meta-analysis from Marjanovic et al. [14] reported similar efficacy of high-flow nasal cannula oxygen therapy as compared to NIV. NIV treatment might be considered as safe and effective in AHF settings in addition to standard-of-care, but no significant impact on mortality has been definitely observed [15]. The reduction in venous return, right ventricle preload, and consequentially, on the left ventricle preload, as well as the amelioration in pulmonary vascular resistance, reduction in systemic blood pressure and left ventricle afterload are all features of NIV which contribute to the impact on the symptoms and clinical signs of AHF [13]. For these reasons, the literature describes increasing rates in the early use of NIV in the acute setting [3,6], the elderly being the most representative patients in the NIV treatment group (about 20% in the West Tokyo Heart Failure [WET-HF] registry [16]). Similarly, about 40% of our patients were on NIV. As outlined in Table 2, predictors of NIV were de novo AHF, respiratory failure, NYHA class impairment, and administration of higher IV furosemide doses in this group. Therefore, worsening clinical status of patients suffering with AHF seems to drive the need for NIV in our population. We already demonstrated that lower partial pressure in oxygen (PaO_2_ ≤ 69.7 mmHg) negatively impacts on the prognosis of patients with AHF [17], thus corroborating the need for counteracting hypoxia via specific ventilation support.

Indeed, the contrasting results of the literature did not provide definite data about the impact of NIV in the acute setting. Gorlicki et al. [18] found no significant reduction in primary endpoint—a composite of in-hospital mortality and 30-day post-discharge death, readmission to hospital, or return visit to the emergency department due to AHF—in those patients with AHF who underwent pre-hospital initiation of NIV. On the opposite side, Carrillo-Aleman et al. [19] outlined a 69% and 61% increase in hospital mortality and 1-year mortality in 300 patients with acute cardiogenic shock who underwent invasive mechanical ventilation, as compared to those on NIV. Similarly, data from the EAHFE (Epidemiology of Acute Heart Failure in Emergency Department) Registry identified a 2.2-fold increase in 30-day mortality in patients on NIV, although the propensity score matching failed in providing significant differences [20]. Specifically, acute coronary syndrome and low systolic blood pressure (SBP < 100 mmHg) were the main determinants of AHF and worse prognosis. These results were independent from the type of HF. In line with our analysis, Metkus et al. [21] effectively demonstrated that the use of NIV still remains associated with a 2-fold higher risk of in-hospital mortality in acute decompensation regardless of LVEF.

Far from recommending the avoidance of NIV in the acute setting of heart failure, the main findings from this study suggest that NIV is not merely a treatment modality but rather a marker for disease severity in AHF.

Kaneko et al. [22] included NIV in their ACUTE HF score for the prediction of a worse outcome in acute HF patients, thus advertising the need for considering NIV as a risk marker in AHF. Our study confirmed the role of NIV treatment as a marker risk both for in-hospital mortality and 180-day follow-up after discharge, although its performance and risk ratio slightly declined over time (Figure 4B). The NIV treatment was associated with a six-fold higher risk for in-hospital mortality and a two-fold higher risk for mortality at 180-day follow-up. The data was independent from BNP and BUN.

The need for better stratifying fora those patients who are admitted to the emergency department and/or the Intensive Care Unit with a diagnosis of AHF is reasonable when considering the overall management. This concept is better derived from the evaluation of Figure 3 and Table 3: NIV use was related to a 4-day increase in LOS, even after adjusting for confounding factors. The Cochrane analysis from Berbenetz et al. [8] did not report any improvement in LOS when non-invasive positive pressure ventilation [NPPV] (CPAP or bilevel NPPV) was applied in patients with acute cardiogenic shock, with the exception of those with PaCO_2_ < 45mmHg, who best benefitted from NPPV. Metkus et al. [6] reported a mean LOS of 5.8 days for patients on NIV, which was higher than those who did not undergo such a treatment (4.8 days, *p* < 0.001) but lower than invasive mechanical ventilation (8.8 days, *p* < 0.001). Miró et al. [9] equally demonstrated no improvement in the short-term prognosis of patients on NIV in the emergency department when presenting with AHF, while underlining the prolongation in hospital stay (+44% increased risk in LOS prolongation). Analyses of 2785 patients with acute HF who were included in the WET-HF registry [20] also found prolonged LOS in patients on NIV.

## 5. Limitations

This research article presents some limitations. First, the retrospective nature of the paper may be considered a potential weakness of the study, along with the single-centre experience. Second, no data were available regarding the mean temporal duration of NIV and its impact on prognosis. Third, the role of disease-modifying therapies at discharge, as well as the influence of specific comorbidities and adverse events (such as acute kidney injury or pneumonia occurrence), might represent potential confounding factors in the final analyses. Further insights should be considered in the near future to refine the results.

## 6. Conclusions

The use of NIV treatment in patient hospitalized for AHF might be associated with worse prognosis and increased duration in LOS. Therefore, the need for NIV application in those patients could be considered as a simple and independent risk marker to predict mortality and intensify the treatments. Despite the retrospective nature of this study, this study enriches the understanding of NIV in the AHF context, emphasizing its dual role as a therapeutic and prognostic tool. While it highlights concerning outcomes associated with NIV, these findings may inform clinical strategies to improve care for high-risk AHF patients.

## Figures and Tables

**Figure 1 biomedicines-13-01844-f001:**
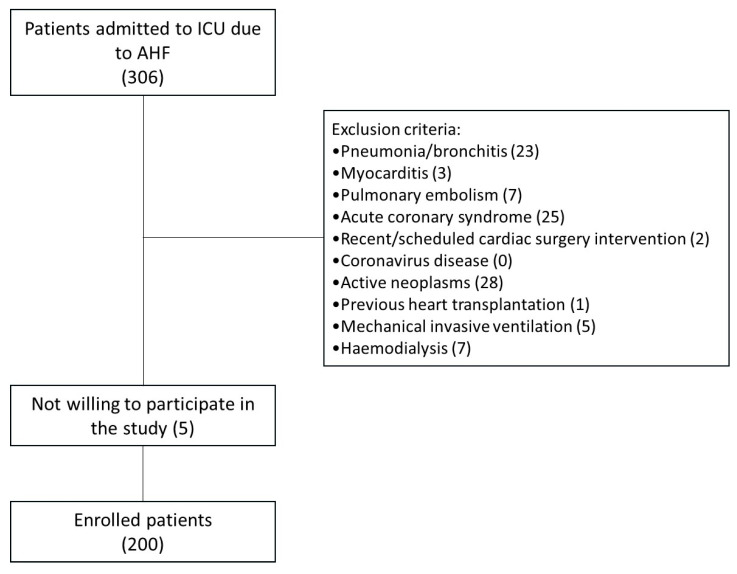
Flow chart of the study.

**Figure 2 biomedicines-13-01844-f002:**
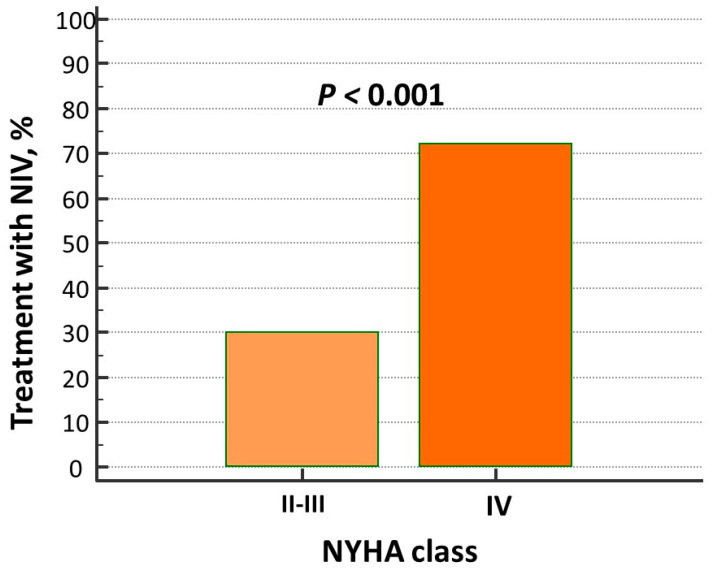
Percentage of patients with advanced NYHA class in non-invasive ventilation (NIV) treatment group.

**Figure 3 biomedicines-13-01844-f003:**
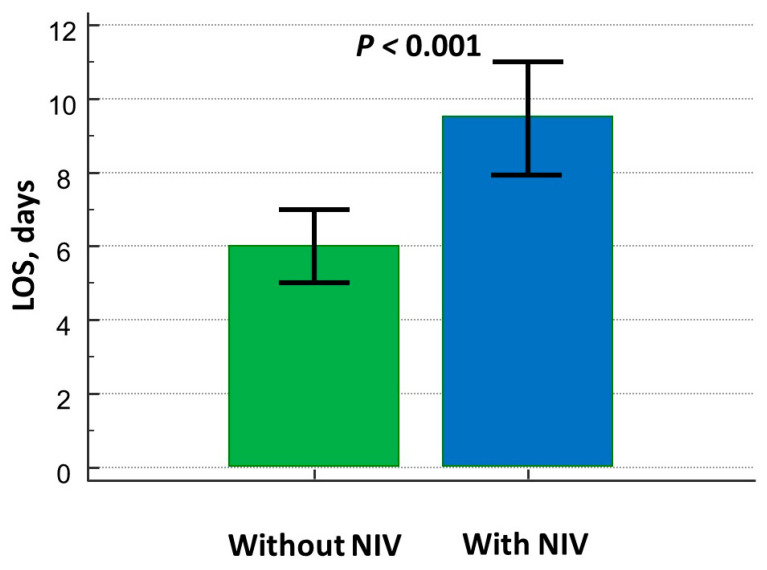
Differences in length of hospital stay between patients on and without NIV treatment.

**Figure 4 biomedicines-13-01844-f004:**
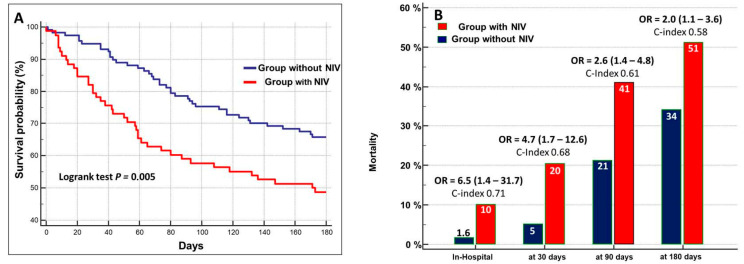
(**A**) Kaplan–Meier survival curves stratified in relation to the administration of NIV. (**B**) Mortality rate in patients treated with NIV as compared to those who did not receive NIV at different times of follow-up. OR = odds ratio (95% confidences interval).

**Table 1 biomedicines-13-01844-t001:** Patient characteristics of study population and differences between patients with and without NIV.

	Overall (*n* = 200)	Non-NIV Group (*n* = 120)	NIV Group (*n* = 80)	*p* Level
**Clinical characteristics**				
Age, yrs	81 ± 9	80 ± 9	81 ± 9	0.3
Male, %	52	55	47	0.3
Body mass index, kg/m^2^	27 ± 6	27 ± 5	28 ± 6	
De novo %	57	50	67	**0.01**
Peripheral edema, %	62	62	63	0.7
Pulmonary crackles, %	63	55	72	**0.02**
Jugular venous distention, %	8	6	10	0.4
Systolic blood pressure, mmHg	127 ± 21	126 ± 22	127 ± 18	0.8
Diastolic blood pressure, mmHg	72 ± 15	72 ± 16	7 1± 14	0.6
Heart rate, bpm	93 ± 27	91 ± 26	94 ± 27	0.3
**Medical history**, %				
Coronary artery disease	33	29	36	0.3
Diabetes	33	30	36	0.4
Atrial fibrillation	43	45	40	0.1
COPD	41	40	44	0.6
PM	12	10	8	0.5
ICD	17	18	7	0.06
**Laboratory values**				
LVEF, %	40 ± 12	40 ± 12	41 ± 12	0.5
Preserved EF	36	35	36	0.9
s-PAP, mmHg	43 ± 16	42 ± 15	45 ± 16	0.2
BNP, pg/ml	1119 (916–1230)	1132 (900–1307)	1115 (849–1302)	0.8
Hemoglobin, g/dL	12 ± 2	12 ± 2	12 ± 2	0.2
BUN, mg/dL	43 ± 28	39 ± 21	49 ± 35	**0.02**
Creatinine, mg/dL	1.7 ± 2.2	1.5 ± 0.8	1.7 ± 1.1	0.07
eGFR, ml/min/1.73 m^2^	52 ± 25	54 ± 22	50 ± 28	0.3
Sodium, mmol/L	139 ± 5	139 ± 8	139 ± 6	0.8
Potassium, mmol/L	4.1 ± 0.6	4.1 ± 0.6	4.2 ± 0.7	0.1
hsCRP, mg/L	13 (11–14)	11 (8–13)	19 (11–40)	**0.001**
Oxygen blood saturation, %	90 ± 5	92 ± 4	88 ±11	**<0.001**
pO_2_, mmHg	64 ± 12	67 ± 12	61 ±11	**<0.001**
pCO_2_, mmHg	38 ± 10	35 ± 6	41 ±11	**<0.001**
Hydration index, %	81 ± 6	80 ± 5	82 ± 8	0.08
e PVS, dL/g	5.4 ± 1.7	5.3 ± 1.7	5.6 ± 1.7	0.2
**Therapies**				
IV Furosemide, mg	120 (60–150)	80 (40–125)	205 (80–250)	**0.001**
Beta-blockers, %	78	77	80	0.6
ACE inhibitors, %	21	45	44	0.5
ARB, %	24	25	22	0.7
ARNI, %	7	8	5	0.4
MRA, %	77	75	78	0.6
SGLT2i, %	10	9	10	0.8
Digitalis, %	11	10	11	0.7
Ivabradine, %	1	1	1	0.8
IV inotropes, %	35	32	40	0.2

Abbreviations: ACE: angiotensin-converting enzyme; ARB: angiotensin receptor blocker; ARNI: angiotensin receptor II blocker–neprilysin inhibitor; BNP: brain natriuretic peptide; BUN: blood urea nitrogen; COPD: chronic obstructive pulmonary disease; eGFR: estimated glomerular filtration rate; ePVS: estimated plasma volume status; hsCRP: high-sensitivity C-reactive protein; ICD: implanted cardioverter/defibrillator; IV: intravenous; LVEF: left ventricular ejection fraction; MRA mineralocorticoid receptor antagonist; NIV: non-invasive ventilation; PM: pacemaker; s-PAP: systolic pulmonary arterial pressure; pCO_2_: partial pressure in carbon dioxide; pO_2_: partial pressure in Oxygen; SGLT2i: inhibitor of the SLGT2 receptor; yrs: years.

**Table 2 biomedicines-13-01844-t002:** Predictors of NIV use in logistic multivariate regression analysis.

Variables	Odds Ratio (95% CI)	*p*	*B* Coefficient	SE	Wald
**De novo vs. CHF decompensated**	2.8 (1.2–6.6)	**0.02**	1.03	0.4	5.6
**Pulmonary crackles, yes vs. no**	1.1 (0.5–2.5)	0.8	0.1	0.4	0.05
**NYHA, II** **–** **III vs. IV**	2.3 (1.1–4.6)	**0.01**	0.8	0.4	6.1
**BUN, × 10 mg/dL**	1.01 (0.99–1.02)	0.1	0.01	0.01	2.4
**hsCRP, × 10 mg/L** **BNP, pg/m** **L** **× 100** **eGFR** **, m** **L** **/min**	1.05 (0.99–1.01)	0.1	0.05	0.03	2.4
**Oxygen blood saturation, %**	0.8 (0.7–0.9)	**0.01**	−0.2	0.08	4.8
**pO_2_, mmHg**	1.01 (0.95–1.07)	0.8	0.03	0.03	0.2
**pCO_2_, mmHg**	1.09 (1.04–1.14)	**0.0001**	0.09	0.02	14.9
**IV Furosemide, × 10 mg**	1.03 (1.01–1.06)	**0.02**	0.03	0.01	5.6

Abbreviations: BUN: blood urea nitrogen; CHF: chronic heart failure; CI: confidential interval; hsCRP: high-sensitivity C-reactive protein; IV: intravenous; NYHA: New York Heart Association class; pCO_2_: partial pressure in carbon dioxide; pO_2_: partial pressure in oxygen.

**Table 3 biomedicines-13-01844-t003:** Predictors of LOS in multivariate regression analysis.

Variables	r	*p*	VIF
**De novo AHF vs. ADCHF**	0.03	0.7	1.06
**NYHA, II–III vs. IV**	−0.02	0.7	1.12
**Oxygen blood saturation, %**	0.11	0.1	1.20
**pCO_2_, mmHg**	−0.004	0.9	1.22
**IV Furosemide, × 10 mg**	0.21	**0.005**	1.14
**NIV group vs. No NIV group**	0.26	**0.0007**	1.4

Abbreviations. ADCHF: acute decompensation of chronic heart failure; AHF: acute heart failure; NIV: non-invasive ventilation; NYHA: New York Heart Association class; pCO_2_: partial pressure in carbon dioxide; LOS: length of stay; IV: intravenous; VIF: variance inflation factor.

**Table 4 biomedicines-13-01844-t004:** Univariate and multivariate Cox proportional hazards survival analyses at 180 days.

	Univariate Cox Regression Analysis		Adjusted Cox Regression Analysis		
	HR (95% CI)	*p*	HR (95% CI)	*p*	Wald
**NIV+ vs. NIV−**	1.84 (1.18–2.85)	0.006	1.7 (1.1–2.9)	0.01	6.0
**Age, year**	1.01 (0.98–1.03)	0.3			
**LVEF, %**	0.98 (0.97–1.01)	0.2			
**NYHA II–III vs. IV**	1.21 (0.77–1.9)	0.4			
**eGFR, ml/min**	0.98 (0.97–0.99)	0.005			
**BUN, × 10 mg/dL**	1.06 (1.03–1.08)	0.0001	1.05 (1.02–1.08)	0.002	9.8
**BNP, × 100 pg/ml**	1.11 (1.07–1.15)	<0.0001	1.02 (1.0–1.04)	0.007	7.3
**hsCRP, × 10 mg/L**	1.04 (1.01–1.07)	0.009			
**pO_2_, mmHg**	1.00 (0.98–1.01)	0.9			
**pCO_2_, mmHg**	1.00 (0.98–1.02)	0.4			
**Oxygen saturation, %**	0.97 (0.93–1.00)	0.1			

Abbreviations: BNP: brain natriuretic peptide; BUN: blood urea nitrogen; CI: 95% confidence interval; eGFR: estimated glomerular filtration rate; HR: hazard ratio; hsCRP: high-sensitivity C-reactive protein; LVEF: left ventricular ejection fraction; NIV: non-invasive ventilation; pCO_2_: partial pressure in carbon dioxide; pO_2_: partial pressure in oxygen.

## Data Availability

Data will be available on request by formal request to the corresponding author.

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
