# Peer review of "The Prognostic Value of Non-Invasive Ventilation in Patients with Acute Heart Failure"

_biomedicines, 2025, doi:10.3390/biomedicines13081844_

Round 1
Reviewer 1 Report
Comments and Suggestions for Authors
In this study, the authors evaluate the role of noninvasive mechanical ventilation in the prognosis of patients with acute HF, specifically after hospital discharge in terms of short-term mortality. This is an important analysis given the high morbidity and mortality in this group of patients.
Overall, the study is well structured and written. I only have a few questions:
1. What was the mean and duration of NIV? Does this influence the outcomes?
2. Are the results in Table 1 from hospital admission?
3. Was the regression in Table 3 a linear regression?
4. Were any in-hospital complications observed in the patients who received NIV? Pneumonia? Thromboembolism? Worsening renal function?
5. What were the percentages of disease-modifying therapy prescriptions at discharge? Could this have been a factor in the outcomes?
6. You should mention the limitations of your study
Author Response
In this study, the authors evaluate the role of noninvasive mechanical ventilation in the prognosis of patients with acute HF, specifically after hospital discharge in terms of short-term mortality. This is an important analysis given the high morbidity and mortality in this group of patients. Overall, the study is well structured and written. I only have a few questions:
- What was the mean and duration of NIV? Does this influence the outcomes?
We would like to really thank the reviewer for this insight. We have no data about the duration of NIV but we do believe that this is a fundamental question. The literature is scant about this specific comment and we regret that we did not previously focused on this issue when performing the original database. We would improve the analyses in the next future by implementing the dataset.
- Are the results in Table 1 from hospital admission?
Yes, the results from table 1 derived from data of hospital admission.
- Was the regression in Table 3 a linear regression?
Yes, the regression in Table 3 is a linear regression.
- Were any in-hospital complications observed in the patients who received NIV? Pneumonia? Thromboembolism? Worsening renal function?
This is a further interesting insight and we would like to really thank the reviewer. We did not report any significant in-hospital complication in our NIV population. For sure, no thromboembolic events were registered for this population. We have not data on the possibility of AKI as itself as we did not include the serial evaluations of kidney function in our database. We can state that no patient underwent haemodialysis or ultrafiltration but we cannot be sure about the occurrence of AKI. Indeed, we have no data about pneumonia occurrence.
- What were the percentages of disease-modifying therapy prescriptions at discharge? Could this have been a factor in the outcomes?
Thank you once again for this comment. We did not include data on disease-modifying therapy prescriptions at discharge. We do agree that these therapies might impact on outcomes and for this reason we will work for upgrading information in the next researches.
- You should mention the limitations of your study.
We included a dedicated paragraph about the limitations of the study.
Reviewer 2 Report
Comments and Suggestions for Authors
Thank you to the editor for the opportunity to review this manuscript.
This study presents a comparison of patient outcomes with acute cardiac decompensation and the need or non-necessity of non-invasive ventilation.
The manuscript is well-structured and easily comprehensible. The methodology is appropriate. The authors cautiously and correctly formulate their conclusions.
We consider the most limited aspect of this work to be its relatively modest medical contribution. The conclusion is clear: patients who were severely ill and required non-invasive ventilation experienced longer hospital stays and poorer outcomes.
Despite the limited clinical impact, the study is methodologically sound, and the conclusions are appropriately drawn. I recommend this manuscript for publication in its current form.
Author Response
Thank you to the editor for the opportunity to review this manuscript. This study presents a comparison of patient outcomes with acute cardiac decompensation and the need or non-necessity of non-invasive ventilation. The manuscript is well-structured and easily comprehensible. The methodology is appropriate. The authors cautiously and correctly formulate their conclusions. We consider the most limited aspect of this work to be its relatively modest medical contribution. The conclusion is clear: patients who were severely ill and required non-invasive ventilation experienced longer hospital stays and poorer outcomes. Despite the limited clinical impact, the study is methodologically sound, and the conclusions are appropriately drawn. I recommend this manuscript for publication in its current form.
We would like to really thank the reviewer for his/her appreciation of our work. Far from being a cornerstone within the complex management of AHF, this paper effectively tried to better specify the role of NIV in this specific setting. Thanks once again for the insights.